# Numerical Study of Aircraft Wake Vortex Evolution under the Influence of Vertical Winds

**Jianhui Yuan [1,2], Jixin Liu [1,2,*], Changcheng Li [1,2]** and **Zheng Zhao [1,2]**

1   College of Civil Aviation, Nanjing University of Aeronautics and Astronautics, Nanjing 211106, China; yuanjianhui@nuaa.edu.cn (J.Y.); lichangcheng@nuaa.edu.cn (C.L.); zheng_zhao@nuaa.edu.cn (Z.Z.)
2   State Key Laboratory of Air Traffic Management System, Nanjing 211106, China
*   Correspondence: larryljx66@nuaa.edu.cn

**Abstract:** Separating wake vortices is crucial for aircraft landing safety and essential to airport operational efficiency. Vertical wind, as a typical atmospheric condition, plays a significant role, and studying the evolution characteristics of wake vortices under this condition is of paramount importance for developing dynamic wake separation systems. In this study, we employed the SST k-ω turbulence model based on an O-Block structured grid to numerically simulate the simplified wing model. We analyzed the variations in the wake vortex structure and parameters of the Airbus A320 during the near-field phase under different vertical wind directions and speeds. The results indicate that favorable vertical winds cause a "flattening" deformation in the wake vortex. Vertical winds reduce the initial vortex strength, accelerate the rate of vortex decay, and influence the trajectory of the vortex core. Notably, under wind speeds of 1~3 m/s, the decay rate is more significant than under 4 m/s. When vertical wind speeds are substantial, it can lead to irregular motion and interactions within the vortex core, forming secondary vortices.

**Keywords:** numerical simulation; wake flow; vertical wind; wake vortex motion



## 1. Introduction

As the number of flights in China continues to increase yearly, many airports are gradually reaching their capacity limits, leading to increasingly severe flight delays. Aircraft taking off and landing on the same runway or on closely spaced parallel runways (where the lateral spacing between runways is less than 760 m) must maintain a minimum safe separation distance to avoid the adverse effects of wake turbulence from preceding aircraft [1]. Therefore, reducing wake turbulence separation intervals has become an important research focus for improving aircraft takeoff and landing efficiency and airport capacity [2,3]. The evolution and decay of aircraft wake vortices are closely related to the surrounding atmospheric conditions. Research has shown that wind disturbances can affect the propagation and dispersion of wake vortices [4]. In particular, vertical wind directly impacts the structure, vortex strength, decay rate, and altitude of the vortex core, which subsequently affects wake turbulence separation and the extent of the danger zone. Therefore, it is essential to conduct detailed research on the evolution and detection of wake turbulence in vertical wind conditions, which is necessary for establishing dynamic wake turbulence separation criteria under different atmospheric backgrounds.

Research methods for aircraft wake turbulence primarily include wind tunnel experiments, laser radar measurements, theoretical modeling, and computational fluid dynamics (CFD) [5].

1.   Wind Tunnel Experiments: Wind tunnel testing is a traditional method of studying aircraft wake turbulence. It involves creating scaled-down physical models of airplanes and observing wake turbulence effects. However, the atmospheric environment's complexity and the measurement equipment resolution limitations restrict this

method's precision. Wind tunnels also have limits when studying the long-distance development of wake vortices [6].

2.  Laser Radar Measurements: Laser radar systems are employed for wake turbulence observations. These systems use lasers to measure various properties of wake vortices. However, their effectiveness can be limited by atmospheric conditions and measurement range [7–9].

3.  Theoretical Modeling: Theoretical models are used to predict the behavior of wake turbulence based on mathematical equations and principles. These models provide valuable insights into wake vortex characteristics, but they may have limitations when capturing all the complex interactions in real-world conditions [5,10].

4.  Computational Fluid Dynamics (CFD): CFD is a computational method that simulates fluid flow based on fundamental physical principles. It offers high scalability, strong visualization capabilities, and the ability to conduct simulations for various aircraft types under different flight phases and atmospheric conditions. In recent years, CFD has found extensive applications in aircraft wake turbulence research and prediction [11,12].

CFD simulations can provide valuable insights into the behavior of wake vortices, their evolution, and the effects of different atmospheric conditions, including wind. This computational approach allows researchers to conduct extensive and detailed studies that can be challenging or impossible to achieve through other methods [13].

Scholars use different turbulence models to numerically simulate the formation of the wake vortex and the evolution of the near-field stage. Crouch et al. simulated the initial formation and subsequent evolution of the wake vortex through the Reynolds-averaged Navier–Stokes (RANS) approach and assessed the safety interval between different models in the terminal area through the intensity of the wake vortex [14]. Meng and colleagues utilized adaptive mesh-based Large Eddy Simulations (LES) to analyze the near-ground evolution of vortices in the wake of the ARJ21 aircraft, initialized using a lift–drag model. Their results showed that the horizontal tail vortex is the weakest and dissipates rapidly, while the wingtip vortex is the strongest and leads to a fusion with the horizontal tail vortex. Furthermore, the comparative analysis suggested that traditional wake vortex models can be applied to study the far-field evolution of wake vortices [15]. Different turbulence models and solver settings of RANS have been previously performed to investigate the accuracy of near-field wingtip vortices. The numerical simulation results show that the vortex core trace can be obtained more accurately when the spatial dispersion in the numerical calculation is larger than the fifth order by comparing the numerical simulation results with the wind tunnel experimental results of Chow [16,17] and colleagues, who conducted a three-dimensional numerical study to investigate the influence of fixed and moving wall boundary conditions on aircraft aerodynamic characteristics and wake vortex development. The results indicate that fixed walls overestimate the ground's viscous effects, leading to a reduction in the magnitude of the wingtip vortex (WTV). Additionally, the secondary vortex flow is induced by the wingtip and horizontal tail vortex (HTV) [12].

Numerous studies by scholars worldwide have explored the effects of different meteorological and ground conditions on the formation and evolution of wake vortices. Zheng and Ash et al. [18] investigated the development of wake vortices under temperature stratification and other crosswind conditions through numerical simulations of non-constant two-dimensional laminar flow. They analyzed the changes in wake vortex trajectories under different weather conditions. Holzäpfel and Steen et al. [19] obtained 288 in situ wake vortex measurements and the respective environmental conditions by various detection systems. They found that side winds and turbulence affect the decay of the wake vortex and that side winds accelerate the decay rate of the downwind vortex. Zhou and colleagues conducted numerical simulations using an Euler–Euler multiphase flow model to investigate the evolution of wake vortices under different rainfall rates. The results indicated that rainfall accelerates the decay of wake vortices, reduces the tendency for velocity distribution within the vortex core to become smooth, alters the wake vortex's de-

scent pattern, and impacts the trajectories and concentration distribution of raindrops [20]. Holzäpfel and Steen et al. proved that the ground effect accelerates the attenuation of the tail vortex by statistically analyzing the actual detection data and gave a model of tail vortex attenuation under the ground effect based on the former model [19]. Wang and colleagues, employing the Open FOAM solver, conducted Large Eddy Simulations (LES) to investigate aircraft wake behavior. The numerical simulation findings revealed the emergence of secondary vortex structures due to interactions between the wake and ground obstacles. These secondary vortices propagate outward along the vortex axis. Furthermore, the dissipation of the wake was found to vary following the distinct shapes of the ground obstacles [21]. Wei et al. have established a tail vortex dissipation model, a motion model, and adopted the mirror vortex method to assess the strength of the ground effect according to the degree of increase in the spacing of the vortex nuclei, which better reflects the influence of the ground effect on the motion and dissipation of the tail vortex [22]. Fred H. Proctor [23] conducted Large Eddy Simulations (LES) to investigate an unusually long-lived wake vortex phenomenon observed in their test program. This behavior is associated with the environmental crosswind's first and second vertical derivatives.

This article aims to study the impact of vertical wind (both in direction and speed) on the structure and characteristic parameters of aircraft wake vortices during the near-field phase. This study intends to enhance our understanding of wake vortex dynamics under varying vertical wind conditions, critical for improving safety and efficiency in aircraft operations. At present, researchers have made significant progress in studying the effects of typical atmospheric conditions such as crosswinds, ground effects, temperature stratification, and rainfall on the evolution and decay of wake vortices. However, more research on the impact of vertical wind environments still needs to be performed. Therefore, this study holds importance in enhancing the calculation and optimization of wake vortex separations under different meteorological conditions. The specific chapter arrangement for this article is as follows.

The first chapter introduces the research background and significance of this article. It provides an overview of the research methods for wake vortices, their advantages and disadvantages, and summarizes the research progress and findings in wake vortex numerical simulations. The chapter also outlines the content covered in subsequent chapters.

In the second chapter, we describe the whole process of numerical simulation on the Fluent platform. It begins by outlining the construction of simplified wing and fluid domain models. The chapter then delves into the structured grid division of the fluid domain, validates the grid independence of the numerical simulation, and determines the appropriate number of grids. It also specifies the N-S equations, turbulence models, and boundary conditions used in the numerical simulations.

In the third chapter, we focus on analyzing the numerical simulation results. It starts by validating the accuracy of the numerical simulations by comparing the results with theoretical models and radar detection data. The chapter then examines the impact of wind direction and speed on wake vortex structure through iso-surface plots of vorticity. It proceeds to provide a quantitative analysis of wake vortex parameters such as vorticity, vortex trajectory, and axial and vertical velocity and describes the phenomenon of secondary vortices.

In the fourth chapter, the main focus is on summarizing the primary research findings of this article.

## 2. Materials and Methods

The overall structure and framework of this chapter are illustrated in Figure 1. In this chapter, the typical medium-sized aircraft, Airbus A320, is selected as the numerical simulation subject, and simplified wing and fluid domain models are constructed. Using the simplified wing model for the study is beneficial to the extraction and analysis of vortex parameters and saves computational resources. The fluid domain is subjected to structured grid division based on the O-block method in ICEM. Grid independence tests are

conducted to determine the appropriate number of grids and grid partitioning parameters. The governing N-S equations, turbulence models, and boundary conditions are selected to reflect real-world situations, where the turbulence model is the SST k-ω turbulence model, a hybrid model of k-ω and k-e. It can simulate the transport and diffusion of vortices in the wake more accurately, and the computational speed is faster than that of the traditional method, which is suitable for calculating the high back-pressure gradient and shear flow. Numerical simulations are performed on the Ansys Fluent2021R1 platform, and investigations are stopped when the parameter errors meet the simulation requirements or after a certain number of iterations.

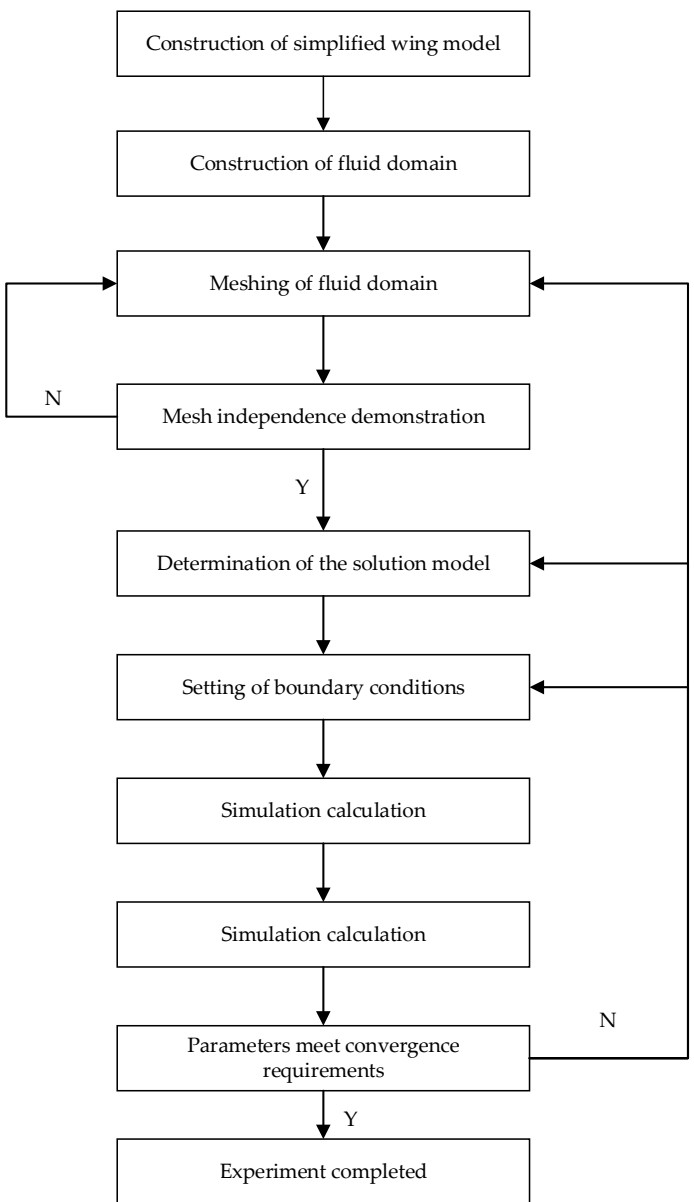

**Figure 1.** Simulation solution process.

### 2.1. A320 Wing Geometry

Figure 2 illustrates the complete model of the Airbus A320 [24] and the simplified wing model used for numerical simulations. There are two primary reasons for this approach. Firstly, the aircraft's trailing vortex is primarily generated at the wingtip, and a smoother wing configuration enhances the strength of the trailing vortex, aiding in the subsequent extraction and analysis of vortex parameters. Secondly, modeling the entire

aircraft and conducting flow field simulations would require a significant number of grids and computational time, with little considerable benefit for the subsequent analysis of the vortex field. Therefore, it was decided to simplify the geometric model of the Airbus A320 [25], retaining only the wing section responsible for generating wingtip vortices. Table 1 presents the wing parameters of the Airbus A320.

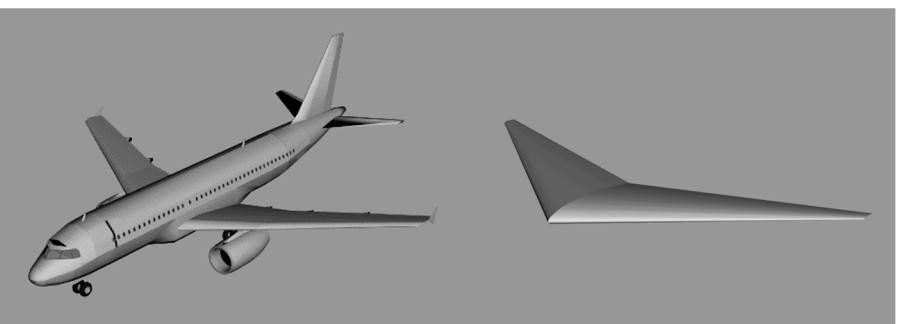

**Figure 2.** Simplified processing model for A320 [26].

**Table 1.** A320 wing geometry parameters [11].

| Geometric Profile Parameters of the Wing | Numerical Value |
| --- | --- |
| Wing span B/m | 36.9 |
| Wing chord length $C_r$/m | 10 |
| Wing area $S_\omega$/m$^2$ | 210 |

### 2.2. Fluid Domain Model

To ensure that the natural approach and landing process of the aircraft can be accurately simulated, shielding the interference of other factors on the aircraft wake, the fluid domain inlet distance from the wing model is set to $4C_r$ ($C_r$ is the wing chord length), and the exit distance from the wing model is $25C_r$, which also effectively mitigates the adverse impact of backflow phenomena on the stability of the numerical solution. The upper wall distance from the wing model is $3C_r$, the lower wall distance from the wing model is $5C_r$, and the distance of the left and right side walls from the wing model is $3C_r$. Figure 3 shows the computational fluid domain model.

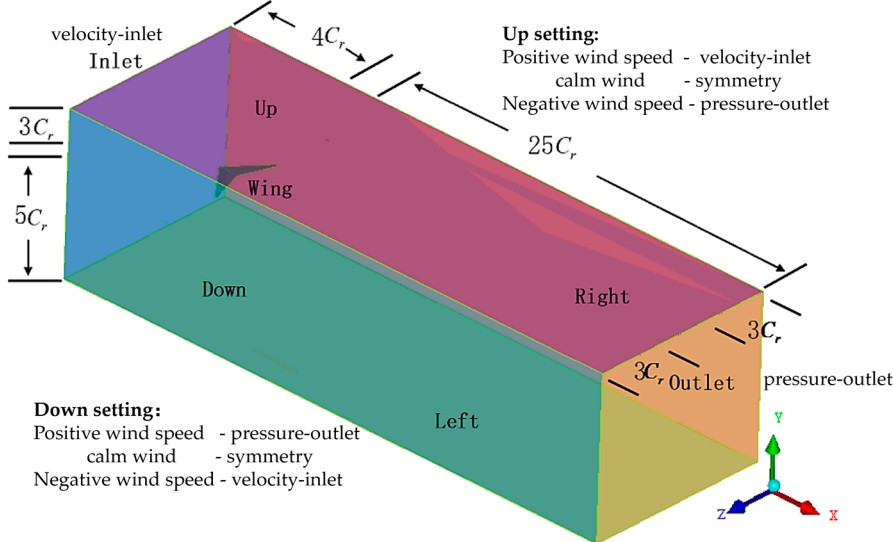

**Figure 3.** Fluid domain model.

### 2.3. SST k-ωModel Governing Equations

The wake vortex flow field is a low-speed flow problem in the aircraft approach phase. Hence, the control equations use the Navier–Stokes (N-S) equations for incompressible fluids with constant viscosity, including the mass conservation equation and momentum equation [27]. The RANS turbulence model is more computationally efficient and stable for complex geometries than DNS and LES, making it suitable for steady flows and practical engineering applications with limited computational resources. The SST k-ω turbulence model has the advantages of high accuracy, high stability in high-Reynolds-number flows, and high computational efficiency, and the model is especially capable of capturing the details of near-wall turbulence, which is very important for the safety and performance of airplanes, and is often used in the RANS method for the study of aircraft wake vortices. Several research articles [11,28–30] have discussed the advantages of the model in detail and justified the choice of the model for the study.

Mass conservation equation:

$$\nabla U = 0, \tag{1}$$

Momentum equation:

$$U \cdot \nabla U = -\frac{\nabla P}{\rho} + \upsilon \nabla^2 U + g - \nabla \cdot (\overline{u'u'}), \tag{2}$$

where $U$ is the velocity vector, $\rho$ is the fluid density, $P$ is the pressure, $v$ is the kinematic viscosity (kinetic viscosity divided by density), g is the volumetric force (e.g., gravity), $\overline{u'u'}$ is the Reynolds stress tensor, which is the second-order moment of the velocity rise and fall.

The SST k-ω model is chosen for the turbulence model. The turbulent kinetic energy k and specific dissipation rate ω can be obtained from the following equations.

$$U \cdot \nabla k = P_k - \beta^* k\omega + \nabla \cdot \left[ \left( \upsilon + \sigma_k \frac{k}{\omega} \right) \nabla k \right], \tag{3}$$

$$U \cdot \nabla \omega = \alpha \frac{P_k}{\upsilon_t} - \beta \omega^2 + \nabla \left[ \left( \upsilon + \sigma_\omega \frac{k}{\omega} \right) \nabla \omega \right] + 2(1 - F_1) \frac{\sigma_{\omega 2}}{\omega} \nabla k \nabla \omega, \tag{4}$$

$P_k$ is the turbulence production term; $\beta^*$, $\alpha$, $\beta$ are the model constants, determined through calibration with empirical or experimental data; $\sigma_k$, $\sigma_\omega$ are the turbulent Prandtl numbers for the diffusion terms; $\upsilon$ is the molecular viscosity; $\upsilon_t$ is the turbulent viscosity, representing the viscous effects of turbulence; $F_1$ is a blending function; $\sigma_{\omega 2}$ is the turbulent Prandtl number for the cross-diffusion term.

Combining the N-S governing equations with the SST k-ω turbulence model ensures the closure of the numerical computation process.

The simulation employs a coupled algorithm and uses a second-order upwind scheme to discretize the governing equations. Convergence is checked by observing residuals, with further iterations added after criteria are met to ensure solution stability.

### 2.4. Meshing and Mesh Independence Demonstration

This chapter performs structured hexahedral meshing of the fluid domain by ICEM-CFD, emphasizing the speed and quality of meshing, and fine-tuning the O-block meshing of the critical airfoil zone domain. Detailed meshing and meshing parameters are given later.. Simulations were solved using the second-order upwind scheme and the SST k-ω turbulence model, and consistent variables were tested for all four mesh sizes to assess the mesh dependence. Grid independence was verified by comparative analysis of velocity profiles at 170 m behind the wing. Detailed simulation experiments are shown in Figures 4 and 5. A grid of 8.91 M provided the best efficiency and accuracy trade-off and was therefore selected for the simulation.

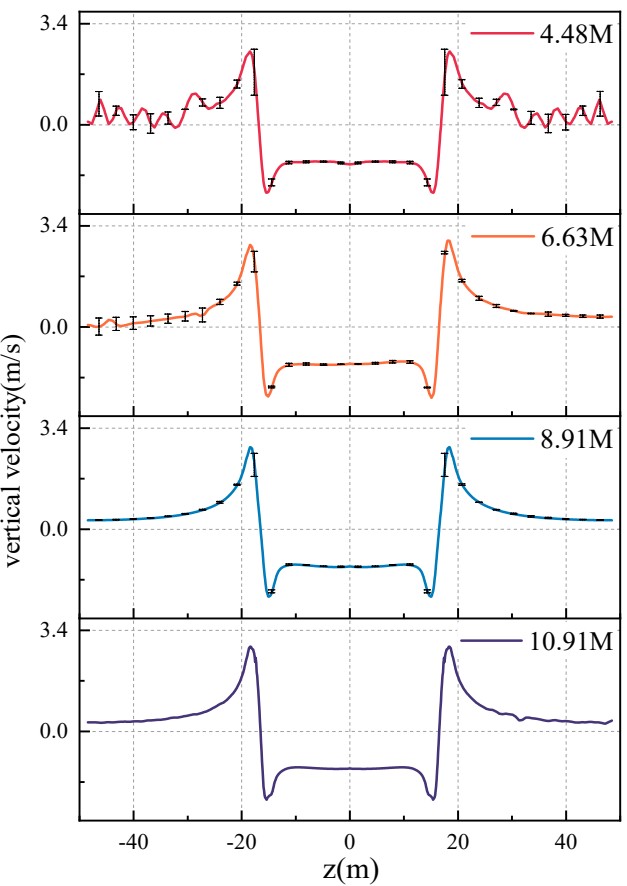

**Figure 4.** The vertical velocity and the error along the vorticities line (x/c = 17).

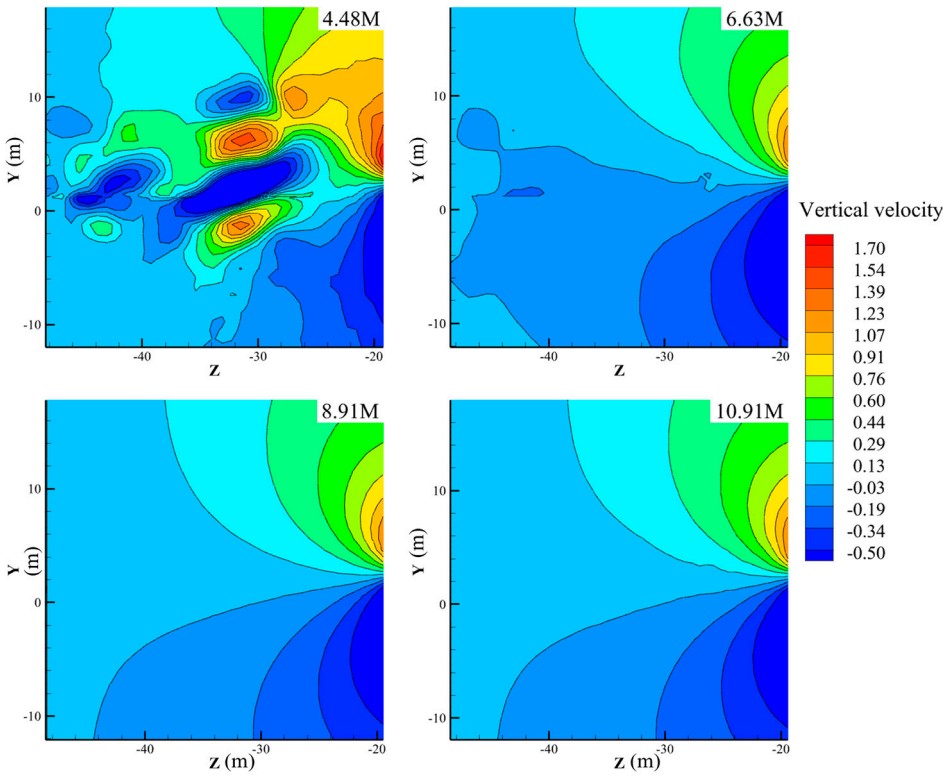

**Figure 5.** Vertical velocity contour at z ≤ −20 position (x/c = 17).

To select an appropriate grid resolution for our numerical simulations, we conducted tests using four different grid sizes: 4.88 million (abbreviated as 4.88 M), 6.63 million (abbreviated as 6.63 M), 8.91 million (abbreviated as 8.91 M), and 10.91 million (abbreviated as 10.91 M) cells. All conditions except for the grid resolution remained consistent. Table 2 presents the numerical values of the environmental parameters used in the numerical simulations. To enhance the accuracy of simulating vortex dynamics, a second-order upwind dissipation scheme and the SST k-ω turbulence model were applied. Vertical velocity along the vortex core line at a position 170 m behind the wing's trailing edge was selected for comparative analysis.

**Table 2.** Numerical simulation environmental parameter configuration.

| Environmental Parameters | Numerical Value |
|---|---|
| Atmospheric pressure (Pa) | 104,103.1 |
| Fluid density (kg/m$^3$) | 1.225 |
| Temperature (K) | 289.15 |
| Free stream velocity (m/s) | 69.45 |
| Angle of divergence of the aircraft (°) | 5 |

Figure 4 illustrates the results for these four grid sizes under the same numerical simulation conditions. The black short line represents the benchmark results obtained with the 10.91 million grid. The error values for the other three grid sizes are also shown. Figure 5 displays the vertical velocity cloud map at *z*-axis coordinates $\leq -20$ m.

The vertical velocity profiles for all four grid sizes exhibit similar trends and numerical values. One notable difference is that the 4.88 M grid shows more significant velocity oscillations and higher errors, occasionally manifesting spurious vortices outside the vortex core region. As the grid resolution increases, the velocity curves become smoother, and the errors decrease. Notably, the 8.91 M grid results in vertical velocity errors, mainly within the range of 0.1 m/s. The vortex core positions and vorticity values are consistent for all four grid sizes. However, the vorticity decays faster in the 4.88 M grid due to its relatively low resolution.

In conclusion, the 4.88 M grid needs to be more coarse, and its simulated flow field around the vortex core needs to be consistent with the actual conditions. The 6.63 M grid shows some improvement but still exhibits higher errors, posing a risk of distortion in long-distance simulations. Both the 8.91 M and 10.91 M grids produce very similar velocity distributions and accurately capture the vortex structure, meeting the grid requirements for experimental simulations. However, the 10.91 M grid demands more computational resources for only a marginal improvement in accuracy. Therefore, we have chosen the 8.91 M grid for further simulation studies.

The structured hexahedral meshing of the fluid domain model is performed using ICEM-CFD. Structured grids offer faster grid generation, higher quality, and better suitability for fluid flow and wing surface tension calculations than unstructured grids. The high curvature at the leading and trailing edges of the wing necessitates the generation of finer grids to ensure computational accuracy. However, simply increasing the number of grid nodes significantly reduces the efficiency of numerical calculations. Therefore, the O-block meshing method is adopted to refine the wing's boundary layer mesh. This method divides the boundary area into several small areas like circles, making the distribution of grid points more uniform and improving accuracy and computational efficiency, especially in the simulation of fluid physical properties. The O-grid can better adapt to the complexity of the geometric structure, so it is more suitable for complex flow simulations such as high-speed flow and turbulence.

The grid height of the attached surface layer y is calculated as [31]:

$$y = \frac{y^+ \mu}{U_\tau \rho}, \tag{5}$$

where μ is the dynamic viscosity of the fluid, $U_\tau$ is the velocity of the fluid flow, and $y^+$ takes the value of 1. The grid height of the attached surface layer y is taken as $2 \times 10^{-3}$ mm in the actual division. The grid quality after partitioning ranges between 0.251 and 1. Figure 6 displays the grid partitions within the computational domain, and Table 3 provides specific details regarding the grid partitioning parameters.

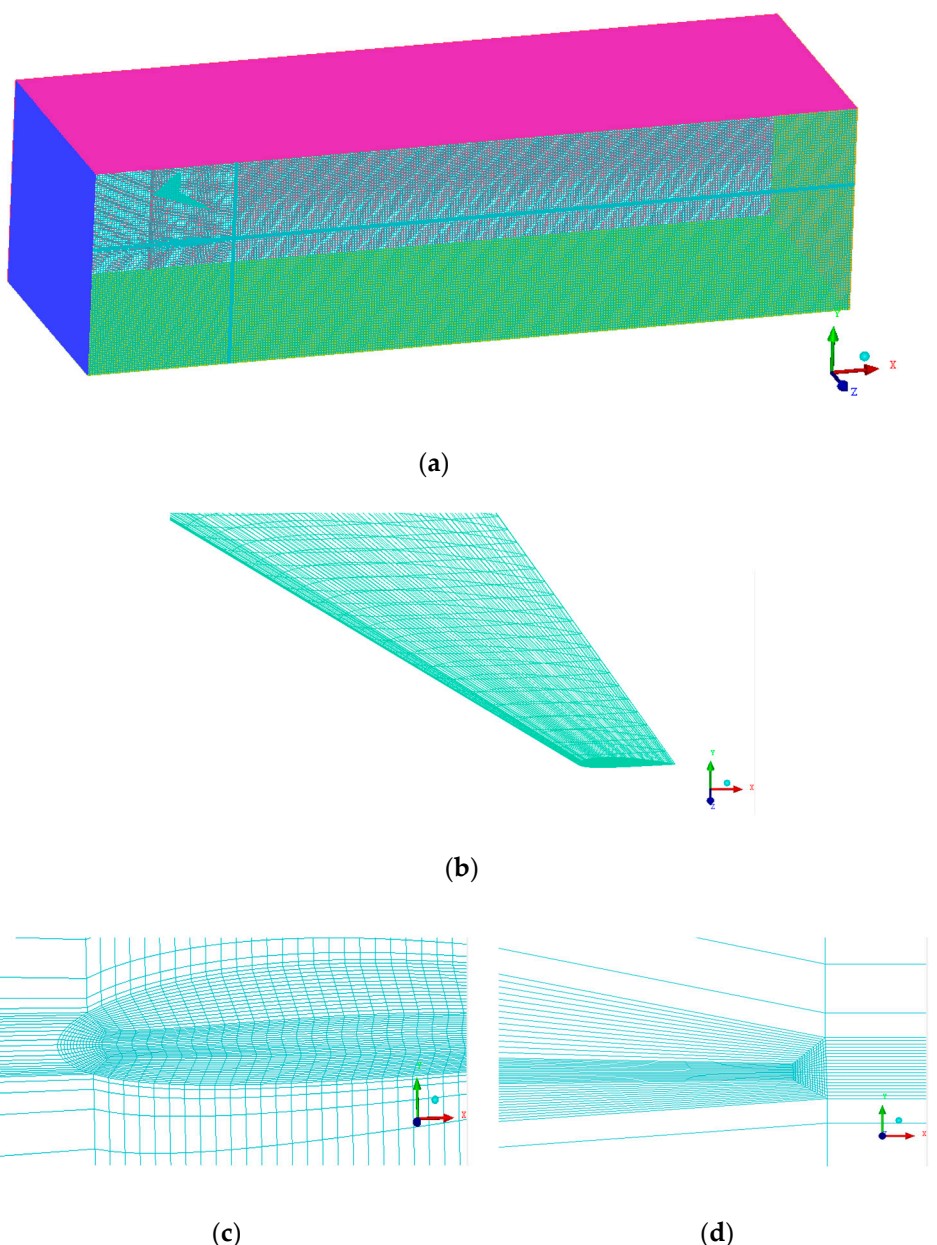

**(a)**

**(b)**

**(c)**          **(d)**

**Figure 6.** Computational fluid domain grid partition. (**a**) Overall; (**b**) wing outer boundary; (**c**) wing leading edge; (**d**) wing trailing edge.

**Table 3.** Grid division parameters.

| Grid Parameters | Numerical Value |
| --- | --- |
| Number of mesh nodes in the wing chord direction | 55 |
| Number of mesh nodes in the wing span direction | 100 |
| Grid growth rate | 1.2 |
| Boundary layer height/mm | $2 \times 10^{-3}$ |

### 2.5. Boundary and Initial Conditions

The boundary conditions assigned to the problem were as follows:

- The inlet velocity is 69.45 m/s, uniform, and the velocity direction is perpendicular to the inlet surface. The incoming velocity of the inlet in the fluid domain is the actual flight speed of the A320 aircraft during the approach phase. Vertical wind conditions were characterized by velocities of −2~4 m/s, uniform velocities, velocity directions perpendicular to the upper and lower surfaces, and the vertical downward direction defined as the positive wind direction. The step size of the simulation experiment was 1 m/s, and seven number-value simulation experiments were conducted. When the wind speed is positive, the upper surface is set as the velocity inlet and the lower surface is set as the pressure outlet; when the wind speed is zero, the upper and lower surfaces are set as the symmetric surfaces; when the wind speed is negative, the lower surface is set as the velocity inlet and the upper surface is set as the pressure outlet, which can be referred to in Figure 3. The values of the wind speeds are based on the actual operating conditions of civil aviation air traffic management.
- Pressure outlet at atmospheric pressure.
- Left and right walls are symmetric surfaces.
- Slip condition on the wing surface.

The solver selects the pressure-based model solver, the finite volume method is used for discretization, the flow field uses ideal gas, the turbulence model sets the SST k-$\omega$ turbulence model, and Table 4 presents the initial conditions at the inlet. All other environmental parameters in the numerical simulations are consistent with those detailed in Table 2.

**Table 4.** Initial conditions at the inlet.

| Variable | Numerical Value |
| --- | --- |
| Free stream velocity | 69.45 m/s |
| Vertical wind speed | −2, −1, 0, 1, 2, 3, 4 m/s |
| Turbulent intensity | 0.5% |
| Turbulent viscosity ratio | 1 |

### 2.6. Simulation Performance

Simulations were performed on a workstation with a 12th Gen Intel®® Core™ i5-12500H, featuring 12 cores and 16 threads at 2.5 GHz, with 32 GB RAM. This leveraged the multi-core architecture for parallel processing, enhancing efficiency and reducing time, providing a suitable alternative to high-performance clusters for our simulation's scope.

Since the numerical simulation cases used in this study have a very high similarity, and the evolution of the tail vortex under different working conditions is investigated, and only some of the boundary conditions and velocity values are adjusted under different vertical wind speeds, batch processing can be used to improve the computation rate, and the relevant code files are contained within the Supplementary Materials. The algorithm selects a coupled algorithm, and the pressure, momentum, turbulent energy equations, and diffusion terms were discretized using the second-order upwind scheme. During the calculation process, we monitor the residual values of the variables to determine whether the convergence is reached. Figure 7 shows the state of convergence of the computed parameters with the number of iteration steps in the simulation calculations.

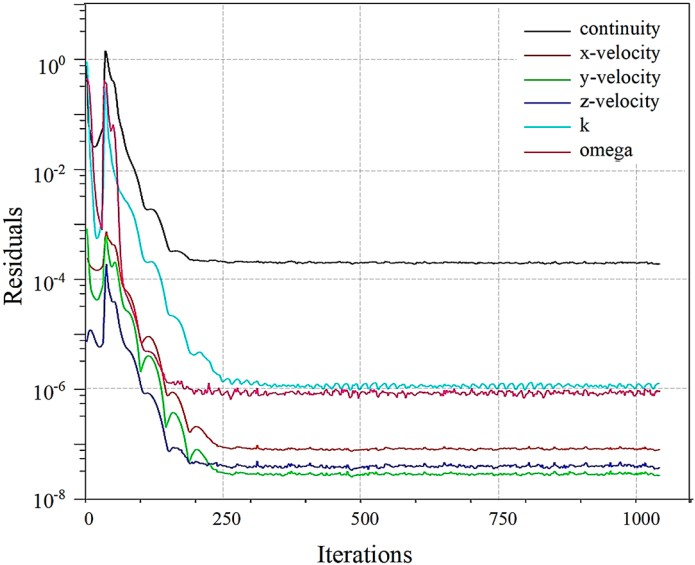

**Figure 7.** Convergence status of computational parameters.

## 3. Results

This chapter primarily focuses on analyzing the downstream flow field resulting from the numerical simulations conducted in Section 2. It involves several key objectives. The numerical simulation results are rigorously compared with the Hallock–Burnham theoretical model and radar data from the existing literature. This comparative analysis is essential for validating the accuracy and reliability of the numerical simulation findings. The chapter then calculates the vorticity distribution at various cross-sectional positions under different wind directions and speeds. This visual representation aids in qualitatively assessing how wind direction and speed impact the structure of the tail vortex. Furthermore, a quantitative analysis investigates how tail vortex parameters evolve as the axial distance increases. This analysis includes parameters such as vorticity, vortex core position, axial velocity, and vertical velocity, and it is performed under varying wind directions and speeds. In addition to these analyses, the chapter explores the occurrence of secondary vortices within the tail vortex.

Through these analytical processes, this chapter provides an in-depth understanding of the numerical simulation results, their accuracy through comparison, and how different wind conditions influence them. This information is crucial for comprehending the behavior of tail vortices under diverse environmental circumstances.

### 3.1. Numerical Simulation Verification

To validate the accuracy of the numerical simulation results, a comparative analysis is conducted among the simulation outcomes under calm wind conditions, radar detection findings from the literature [32], and calculations derived from the Hallock–Burnham theoretical model [24]. The tangential velocity of the vortex at a single point in the Hallock–Burnham model is

$$V_\theta = \frac{r\Gamma}{2\pi(r^2 + r_c^2)},$$ (6)

and the vertical velocity at any point $(x, y)$ in the flow field is

$$V_y = \frac{\Gamma_1(x - x_1)}{2\pi\left[(x - x_1)^2 + (y - y_1)^2 + r_{c1}^2\right]} + \frac{\Gamma_2(x - x_2)}{2\pi\left[(x - x_2)^2 + (y - y_2)^2 + r_{c2}^2\right]},$$ (7)

where $\Gamma_i$ denotes wake vortex circulation and $r_c$ denotes the radius of the vortex core. As can be seen in Figure 8, with the change in radial distance, the tangential velocity shows a trend of increasing and then decreasing; the tangential velocity is the largest at the vortex

core, and the rate on both sides of the vortex core is dropping and tends to zero. Due to the differences in models and measurement locations, the specific values of the velocity are also different; it can be remarkably observed that the velocity distribution patterns obtained from the numerical simulation closely align with both the Hallock–Burnham theoretical model and the radar detection findings.

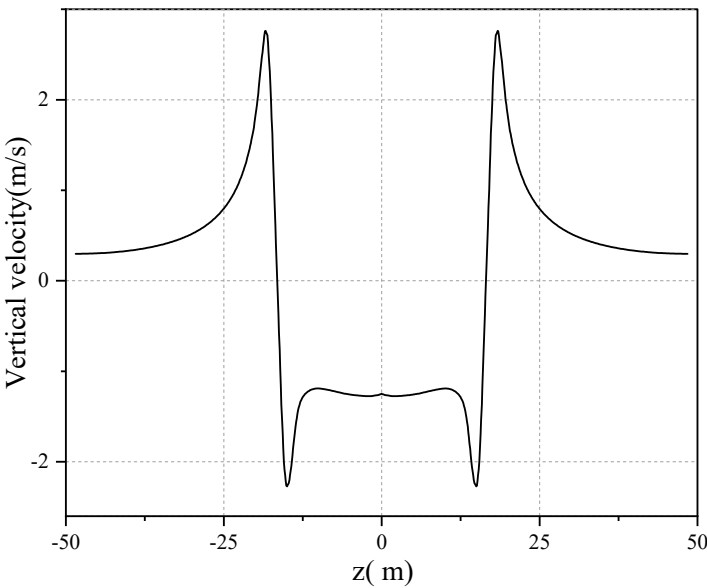

(**a**) Numerical simulation results

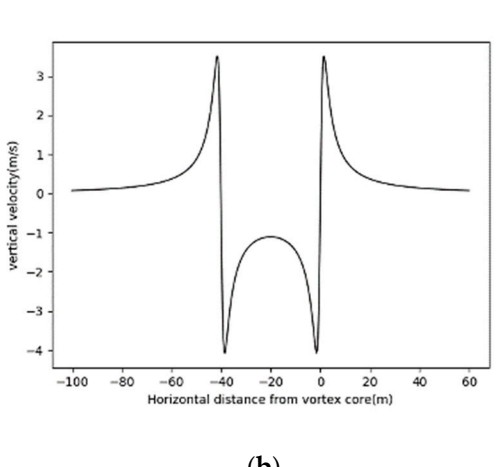

(**b**)

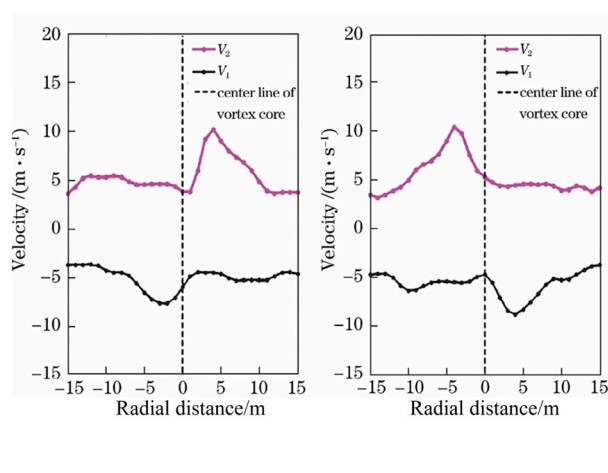

(**c**)

**Figure 8.** Tangential velocity distribution of the wake vortex. (**a**) Numerical simulation results; (**b**) H-B theoretical modeling results; (**c**) radar detection results.

Our study validates its numerical model by comparing results with those from the literature, which investigates the aerodynamics and vortex evolution of A320 under various conditions. Using similar O-H-type structured grids and the SST k-ω turbulence model, our research aligns with the numerical simulation conditions of [12]. The comparison at 60 m height with stationary wall conditions shows great agreement in the wingtip vortices' horizontal and vertical displacements, as illustrated in Figure 9. This consistency confirms the accuracy of our model and bolsters the reliability of our simulation approach for capturing the airflow dynamics around aircraft.

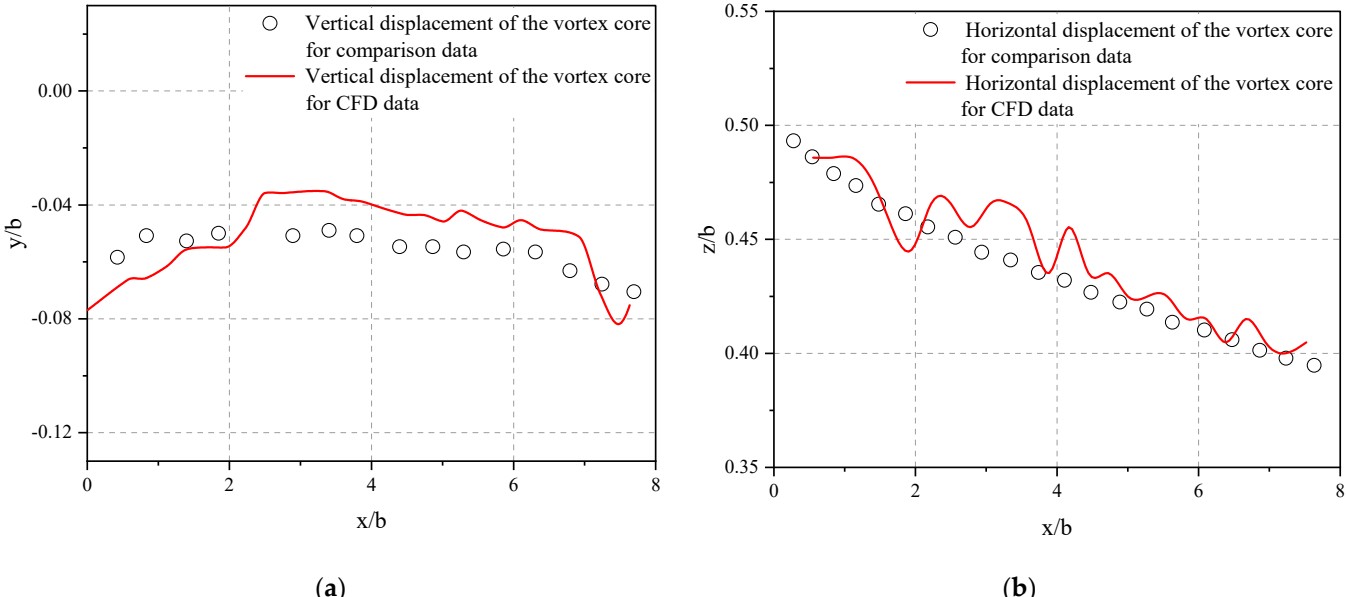

**Figure 9.** Comparison between the results of references and numerical simulations. (**a**) Vertical displacement of the vortex core. (**b**) Horizontal displacement of the vortex core.

### 3.2. Analysis of Tail Vortex Structure

The vorticity magnitude is an important parameter describing the strength of the wake vortex, where the vorticity magnitude $\omega$ is expressed as follows [11]. The maximum vorticity magnitude in the vertical plane is the location of the vortex core, and the distance between the two vortex cores is the vortex core spacing.

$$\omega = \sqrt{\omega_x^2 + \omega_y^2 + \omega_z^2}, \tag{8}$$

$$\omega_x = \frac{\partial w}{\partial y} - \frac{\partial v}{\partial z}, \tag{9}$$

$$\omega_y = \frac{\partial u}{\partial z} - \frac{\partial \omega}{\partial x}, \tag{10}$$

$$\omega_z = \frac{\partial v}{\partial x} - \frac{\partial u}{\partial y}, \tag{11}$$

where $\omega_x$, $\omega_y$ and $\omega_z$, respectively, denote the components of vorticity along the X, Y, and Z axes. The rotational characteristic of the wake vortex is described using the tangential velocity; the expression for the tangential velocity $V_\theta$ is given by:

$$V_\theta = \sqrt{u^2 + v^2}, \tag{12}$$

where $u$, $v$, respectively, denote the component of velocity on the X, Y axis.

A vortex's entire lifecycle can be divided into four distinct stages: Formation Stage, Initial Diffusion Stage, Stable Diffusion and Decay Stage (also known as the mid-to-far-field stage), and Dissipation Stage [33]. Typically, the Formation Stage of a vortex is observed to extend along the wing span for approximately one wing span length. This stage is characterized by differences in lift, causing air to flow downward from around the wingtip to the lower surface, subsequently rolling upward, thereby giving rise to a rotating vortex. During this stage, the vortex exhibits high intensity and is primarily concentrated near the wingtip. Following formation, the vortex enters the Initial Diffusion Stage, spanning a range of approximately 2 to 10 times the wing span length. The vortex still preserves its rotating characteristics during this stage, albeit with diminishing strength and core spacing.

The vortex undergoes a gradual process of stretching and thinning, elongating along the flight direction and reducing its cross-sectional area. The Stable Diffusion and Decay Stage, also referred to as the mid-to-far-field stage, sees the continued diffusion and weakening of the vortex. While rotation persists, the vortex becomes more diffuse, with a further decrease in intensity. This stage extends significantly from the aircraft and can be sustained considerably. The Dissipation Stage marks the eventual complete dissipation of the vortex, where its characteristics become indiscernible from the surrounding airflow.

Figure 10, through a three-dimensional vorticity cloud map, illustrates the near-field vortex evolution of an A320 aircraft under static wind conditions as it extends along the wing by 260 m, equivalent to seven wing spans. At the one wing span position, the vortex core is already well-defined, and there is an apparent trend of separation between the two vortices. Within the two to seven wing span region, the attenuation in vorticity and vortex core spacing is evident.

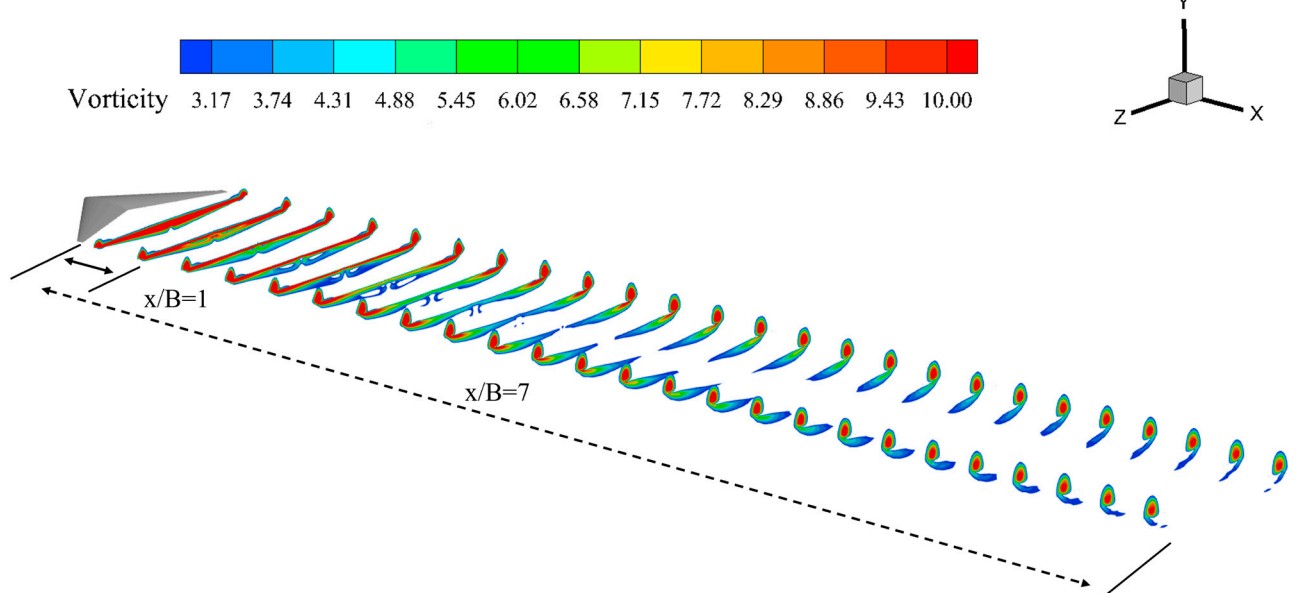

**Figure 10.** Full lifecycle diagram of the vortex under calm air conditions.

Figure 11 shows the vorticity contours at different distances behind the wing under different vertical wind speeds. The statistics show that the airflow will flow from the high-pressure region to the low-pressure area, forming a vortex structure extending backward from the tip of the wing. The vorticity is highest at the vortex core, gradually decreasing outward from the center. The vortex core radius also enlarges with increasing distance from the wing's trailing edge.

Comparing vorticity cloud maps under different wind speeds, it becomes evident that the vortex core position gradually rises under wind conditions of −1 m/s and 0 m/s, and the overall vortex structure remains relatively consistent. However, under −1 m/s wind conditions, there is a faster vorticity decay. At the exact cross-sectional location, the core spacing is smaller, and the complete separation of the two vortices is achieved at an earlier position.

Subsequently, under forward wind speeds of 1 m/s, 2 m/s, and 3 m/s, the vortex structure undergoes noticeable deformation. This is primarily due to the disruptive effects of wind disturbances and turbulence within the wind field, which disturb the internal balance of the vortex. As a result, the vortex shape changes, with distortions such as twisting or stretching, leading to an overall 'flattened' appearance. The most pronounced deformation occurs under 1 m/s wind conditions, and vorticity and core spacing decrease

faster than still air conditions. Additionally, the vertical displacement of the vortex core increases with higher wind speeds.

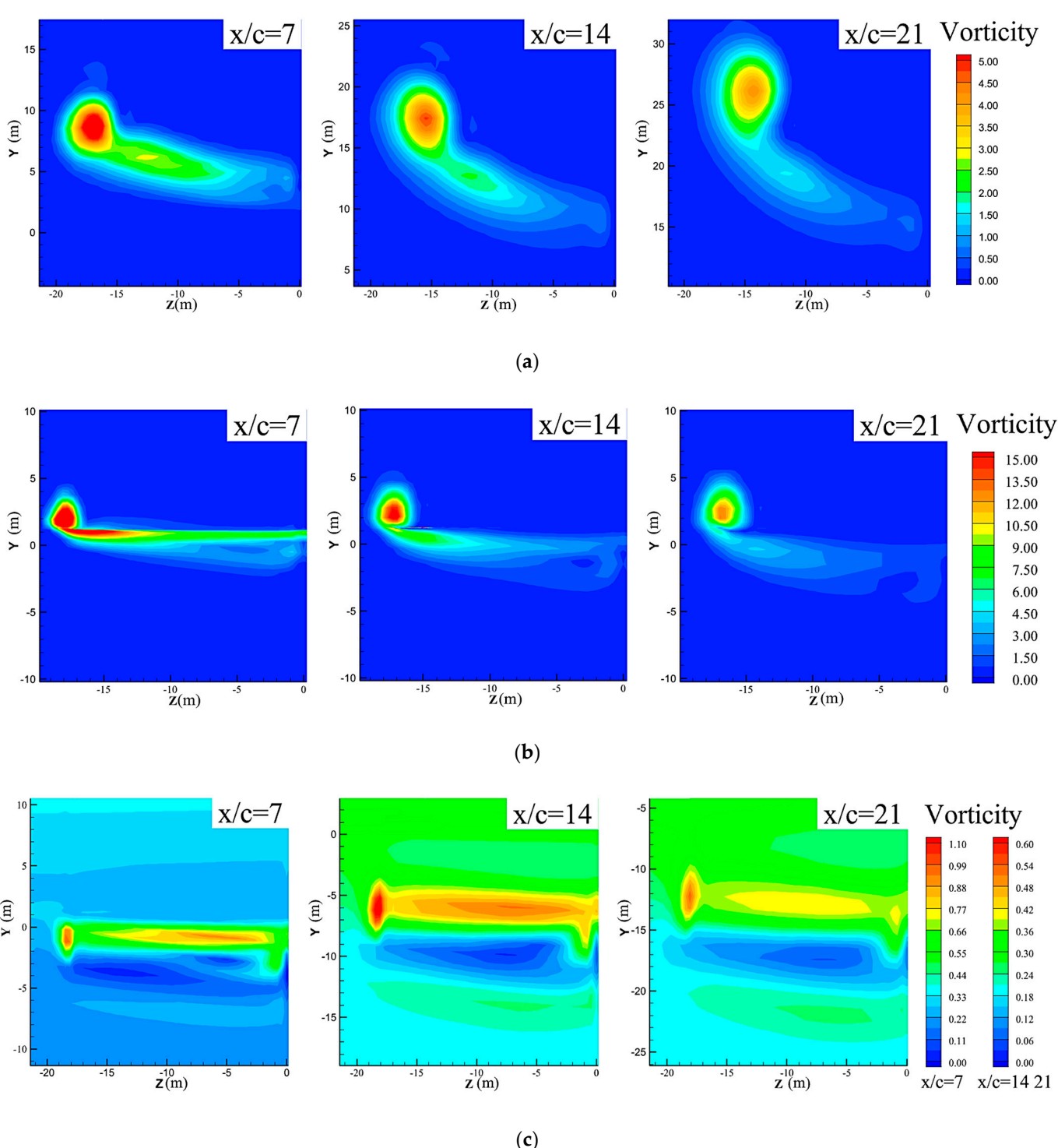

**Figure 11.** *Cont.*

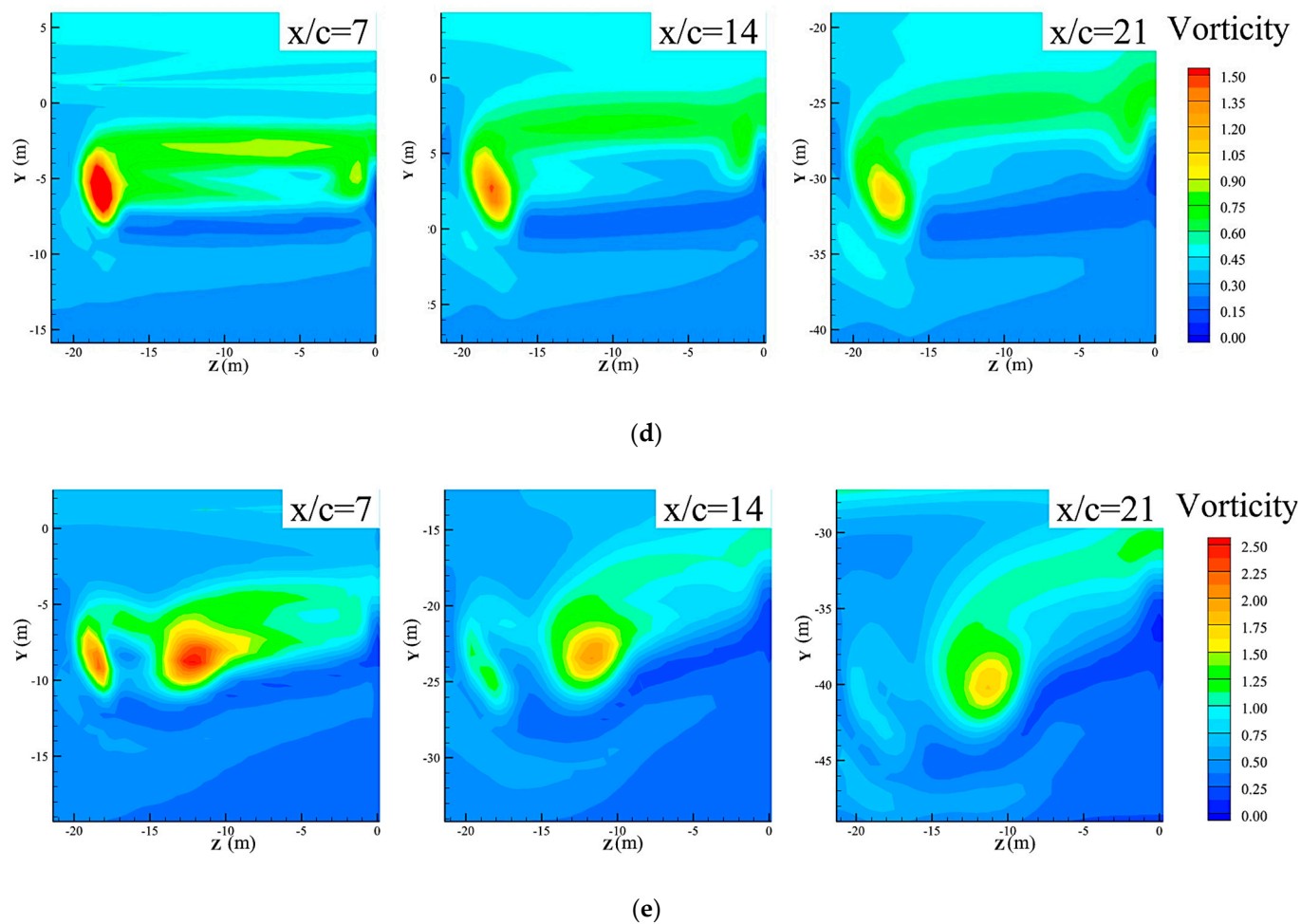

**Figure 11.** Vorticity contour profile of different positions behind the wing. (**a**) Wind speed −1 m/s; (**b**) wind speed 0 m/s; (**c**) wind speed 1 m/s; (**d**) wind speed 2 m/s; (**e**) wind speed 3 m/s.

Notably, under 3 m/s wind conditions, a secondary vortex separates from the primary vortex and attaches to its outer side. With increasing distance of backward movement, the secondary vortex rapidly decays and gradually dissolves into the primary maelstrom. Turbulence, characterized by irregular, rotating motion within the airflow, destabilizes the structure of the primary vortex to a certain extent. Intense turbulence can trigger the generation of secondary vortices.

### 3.3. Analysis of Wake Vortex Parameters

Figure 12 illustrates vorticity distribution at different distances behind the vortex core with varying vertical wind speeds. It can be observed that vorticity rapidly decays in the early stages of vortex formation, with a slower decay rate beyond $x/c = 15$. The influence of vertical wind reduces the initial vorticity of the vortex core and significantly accelerates the rate of vorticity decay. Comparing the two wind direction conditions under the same wind speed magnitude, it is evident that vorticity is consistently higher under negative wind speeds. This is attributed to convective effects, as upward-blowing winds are typically associated with ascending air currents. The ascending air may carry away momentum from the vortex and promote the re-mixing of upper-level air into the vortex, thereby prolonging the lifespan of the vortex.

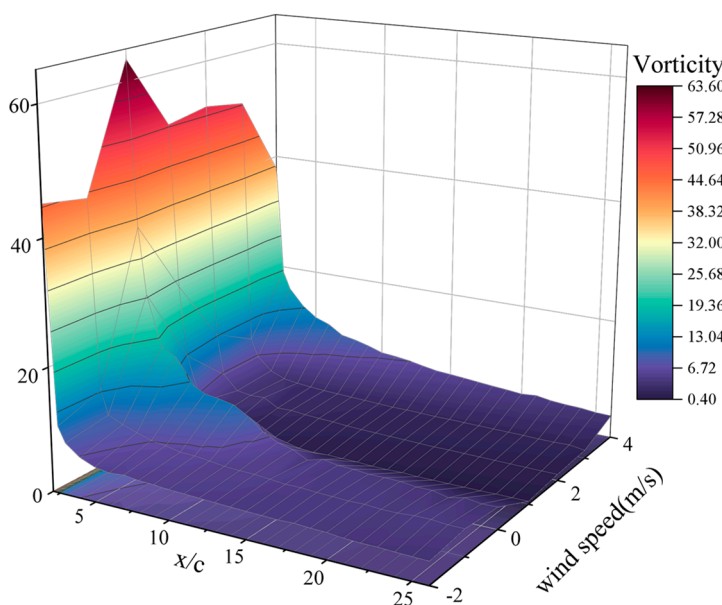

**Figure 12.** Vorticity distribution at vortex core with changing distance at different wind speeds.

Contrary to expectations, there is not a positive correlation between vorticity decay rate and wind speed under favorable wind conditions. Specifically, at the $x/c$ = six cross-sections, vorticity is 15%, 29%, 23%, and 84% under still air conditions for vertical wind speeds of 1-4 m/s. The addition of vertical wind influences the vortex's rotational speed, alters the structure and distribution of the vortex core, reduces the vortex's stability, and leads to rapid vorticity decay. However, vertical wind also, to some extent, replenishes turbulent kinetic energy and compensates for losses incurred by the vortex due to atmospheric viscosity and turbulence effects. As such, the decay rate of vorticity at the vortex core is not positively correlated with wind speed.

Figure 13 presents the distribution of three-dimensional vortex core trajectories under different vertical wind speeds. The two intersecting points on the left and right correspond to the locations of the wingtips. The changes in the vortex core, both in vertical and horizontal displacement, are visually depicted with distance traveled.

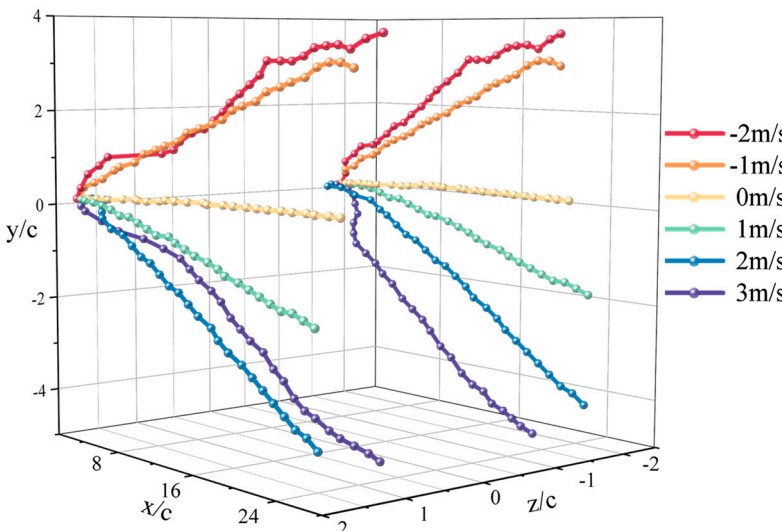

**Figure 13.** Distribution of three-dimensional vortex core trajectories at different wind speeds.

### 3.3.1. Analysis of Wake Vortex Parameters under Different Wind Directions

Figure 14 represents the horizontal position distributions of the vortex core under different positive and negative wind speeds (favorable wind denotes wind blowing from above, while adverse wind denotes wind blowing from below). From Figure 14, it can be observed that under calm wind conditions, the vortex core steadily contracts inwards. Under positive wind speeds of 1 m/s and 2 m/s, the vortex core briefly contracts inwards before stabilizing and oscillating. Specifically, under 2 m/s wind speed conditions, the vortex core spacing is smaller than under 1 m/s wind speed conditions. Under negative wind speeds of 1 m/s, the vortex core steadily contracts inwards at a rate 1.5 times that of still air conditions. At the x/c = 7 cross-section, the horizontal displacement of the vortex core matches that of still air conditions; under negative wind speeds of 2 m/s, the vortex core contracts inward steadily for a distance before rapidly contracting inwards and stabilizing. This phenomenon is known as the detachment of secondary vortices. When wind speeds increase, the airflow kinetic energy within the primary vortex increases. The vortex core may be influenced by wind speed gradients and rotation rate gradients, resulting in the instability of the primary vortex and irregular motion and interaction within the vortex core structure. This can lead to the detachment of the vortex core from the primary vortex and the formation of secondary vortices.

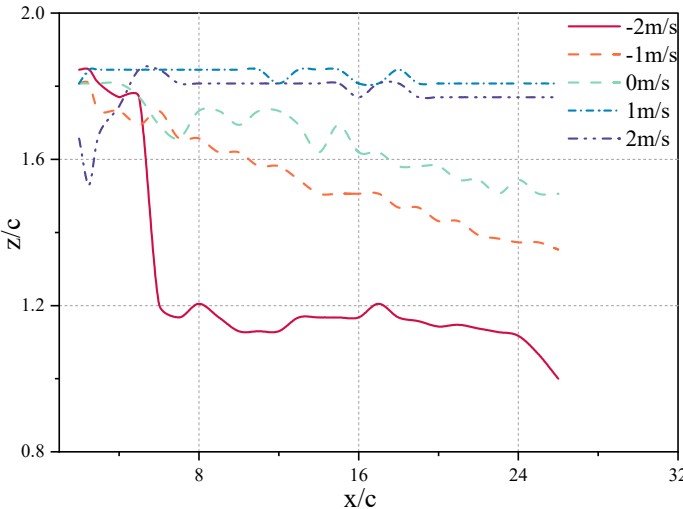

**Figure 14.** Horizontal position distribution of the vortex core under different wind directions.

Figures 15 and 16 display the horizontal position of the secondary vortex core and vorticity cloud maps at different places behind the wing. At the x/c = 5 cross-section, the vorticity of the primary and secondary vortex is consistent. However, the vorticity decay rate and horizontal position change in the two vortices show different trends. The vorticity of the secondary vortex decays more rapidly, and by the x/c = 15 cross-section, it has attached itself to the outside of the primary vortex and merged with it. The horizontal position reduction in the secondary vortex is consistent with the conditions under −1 m/s wind speed.

From Figure 17, it can be observed that under still air conditions, the vortex core exhibits a slight tendency to curl upward, which is in line with the findings of Chow's wind tunnel experiments on near-field wingtip vortices, as documented in the literature. The vortex core tends to move upward and outward in the initial stages relative to the wingtip. This phenomenon is attributed to the "knotting" effect generated by the fusion of the primary vortex and secondary vortices. Under favorable wind speed conditions, the vortex core steadily moves downward, with a rate 2.25 times that of 1 m/s under 2 m/s wind speed conditions. Under adverse wind speed conditions, the vortex core steadily

moves upward, and the absolute value of vertical displacement is roughly consistent with positive wind speeds.

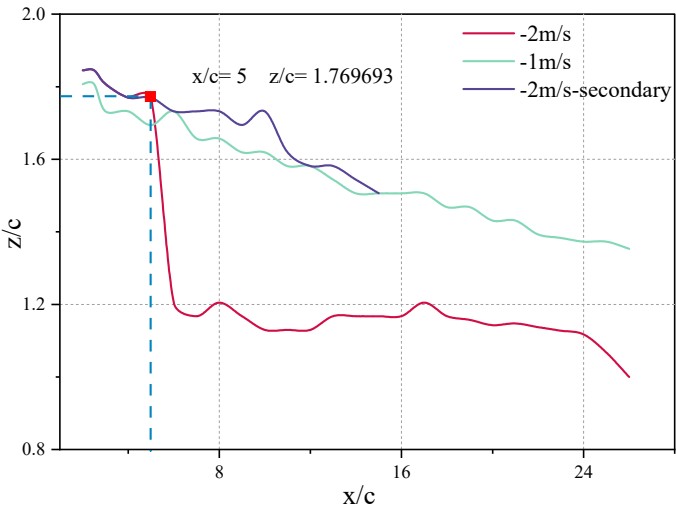

**Figure 15.** Horizontal position distribution of secondary vortex and the primary vortex.

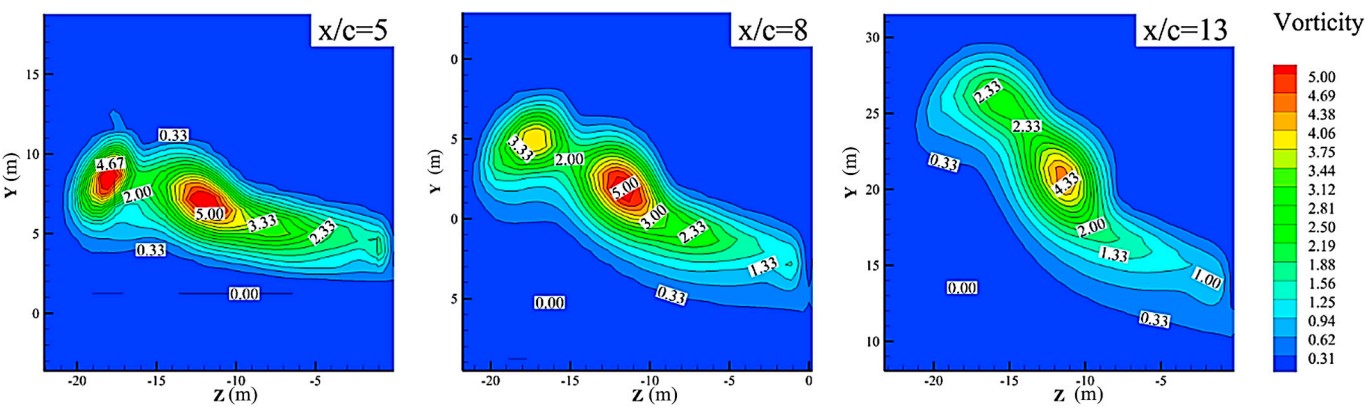

**Figure 16.** Vorticity contour profile of secondary vortex and the primary vortex at −2 m/s wind speed.

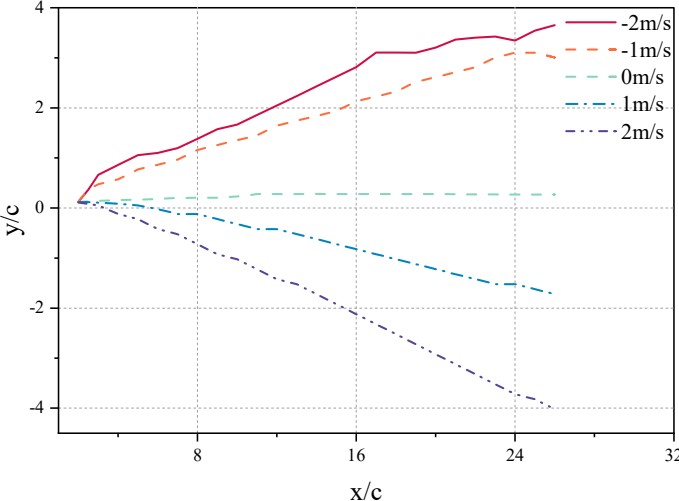

**Figure 17.** Vertical position distribution of the vortex core under different wind directions.

Figure 18 illustrates the vertical and axial velocity distributions at the vortex core under different wind speeds (results are non-dimensionalized using the inflow velocity in the x-direction). The vertical rate at the vortex core fluctuates around −1 m/s. Vertical wind increases the vertical velocity at the vortex core, with a more significant effect as wind speed increases. The absolute value of the rate increases with the downstream distance. Under a wind speed of 4 m/s, at the x/c = 10 cross-section, the vertical velocity at the vortex core reaches nearly 20 m/s. Vertical wind near the vortex interacts with the surrounding air, leading to collisions and momentum exchange. This, in turn, alters the dynamic pressure and enhances turbulence in the flow field near the vortex core, resulting in increased vertical velocity. Under calm wind conditions, the axial speed at the vortex core fluctuates around 70 m/s, maintaining consistency with the aircraft's forward rate. In vertical wind, the axial velocity at the vortex core increases with more incredible wind speeds. At a distance of 200 m behind the wing, under vertical wind conditions of 1 m/s, 2 m/s, 3 m/s, and 4 m/s, the axial velocity at the vortex core increases by 4 m/s, 7 m/s, 8 m/s, and 8 m/s, respectively. This observation agrees with the findings in reference [34], which concluded that crosswinds significantly increase axial velocity in the wake vortex.

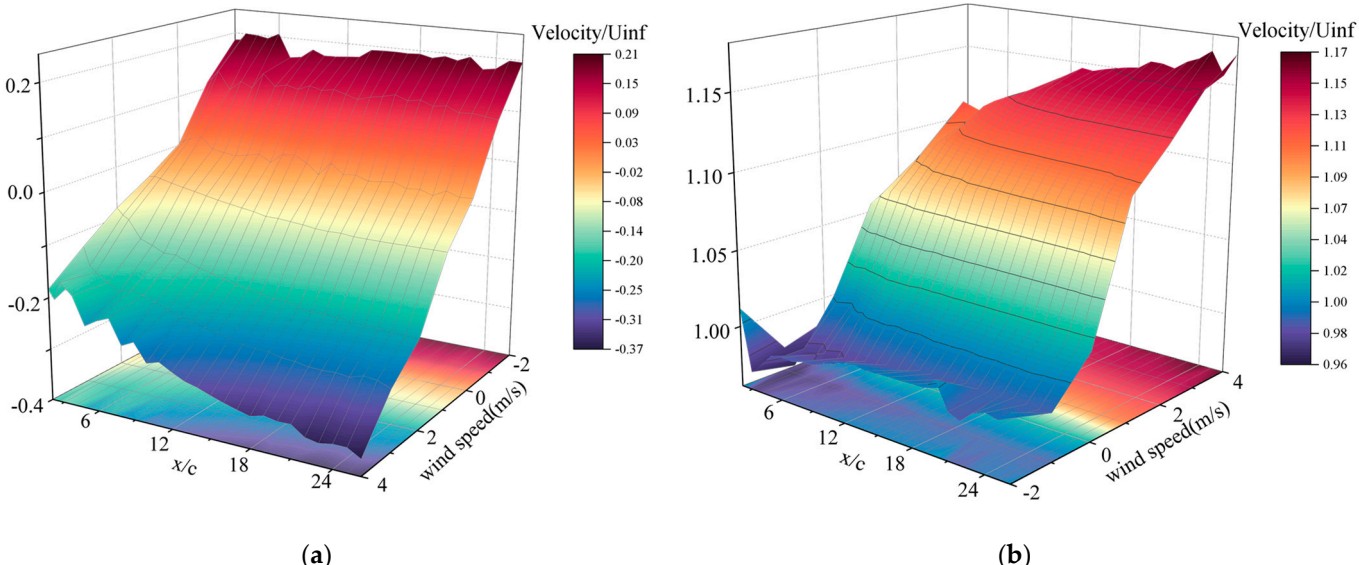

(**a**)                                                                                 (**b**)

**Figure 18.** Vertical velocity and axial velocity of the vortex core at different wind speeds. (**a**) Vertical velocity; (**b**) axial velocity.

3.3.2. Analysis of Wake Vortex Parameters under Different Wind Speeds

Figures 19 and 20 depict the horizontal and vertical displacement distributions of the vortex core under various positive wind speeds, respectively. Figure 18 shows that under wind speeds of 1 m/s and 2 m/s, the vortex core undergoes a brief contraction before stabilizing and oscillating. The vertical wind component restrains the reduction in vortex core spacing. Under wind speeds of 3 m/s and 4 m/s, the vortex core initially contracts rapidly (with contraction nodes at x/c = 4 and x/c = 8) and then steadily contracts at a slower rate. As indicated in Figure 11e, the sudden horizontal displacement of the vortex core results from the detachment of secondary vortices from within the primary vortex due to the instability of its rotation. After the secondary vortices detach, the primary vortex structure stabilizes, and the rate of horizontal displacement reduction becomes steady. Consequently, the changes in vortex horizontal displacement do not exhibit a linear relationship with vertical wind speed. Under wind speeds of 1 m/s and 2 m/s, the vortex core spacing reduction rate is less than that under still air conditions, resulting in a larger downwash danger zone, indicating an increased risk of encountering wake turbulence for following aircraft.

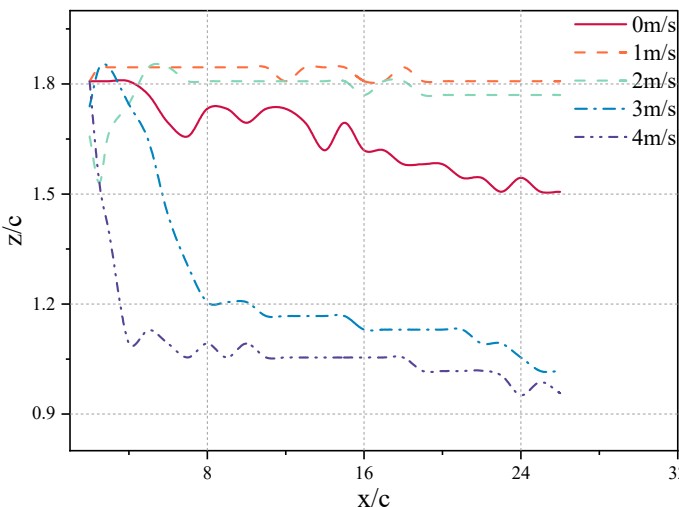

**Figure 19.** Horizontal position distribution of the vortex core at different wind speeds.

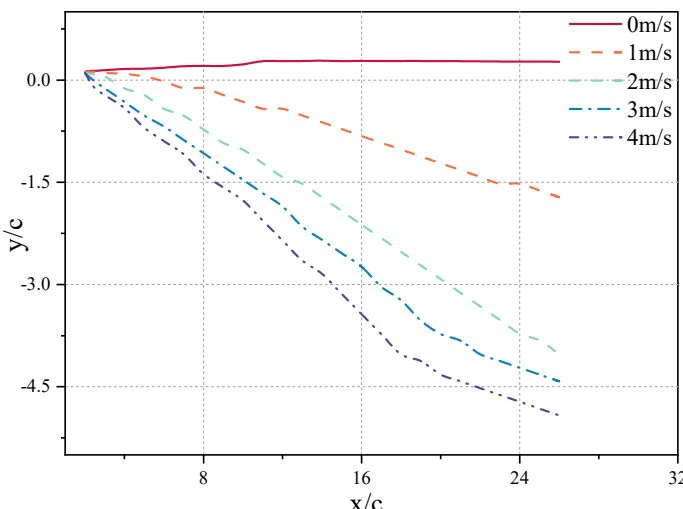

**Figure 20.** Vertical position distribution of the vortex core at different wind speeds.

Figure 20 shows the interaction between wind and the vortex core, leading to the descent of the vortex core through the momentum transfer process. As wind speed increases, the descent displacement of the vortex core also increases. For instance, at the $x/c = 10$ cross-section, the vertical removal of the vortex core under vertical wind conditions of 1 m/s, 2 m/s, 3 m/s, and 4 m/s is approximately 0.08 B, 0.3 B, 0.4 B, and 0.5 B, respectively. Notably, the descent effect on the vortex core is most pronounced under wind speeds of 1 m/s and 2 m/s. Beyond a certain wind speed threshold, the descent effect on the vortex core remains relatively consistent. This may be influenced by factors such as the vortex core's saturation state and the airflow's inertia.

## 4. Conclusions

In this paper, we conducted a study using numerical simulations to investigate the impact of different intensities and directions of vertical wind on aircraft wake vortex structures and vortex parameters, which have provided new insights into the dynamics of aircraft wake vortices under the influence of vertical wind. Different wind intensities and directions can significantly affect vortex structure and decay rates. Negative vertical wind impacts the vortex minimally, maintaining a structure akin to calm conditions, while positive wind alters the vortex significantly. These findings are crucial for devising strategies to

mitigate wake turbulence hazards, especially during the critical phases of flight near the ground, ultimately enhancing aircraft safety and operational efficiency. Specific conclusions were drawn:

1.  Negative vertical wind has minimal impact on the wake vortex structure and remains similar to calm wind conditions. However, favorable vertical wind disrupts the internal balance of the wake vortex and the turbulence within the airflow, leading to distortion or stretching of the vortex structure and causing an overall "flattening" of its shape.

2.  Vortex intensity rapidly decreases during the initial formation of the vortex and then gradually diminishes. Vertical wind reduces the initial vortex intensity and increases the rate of vortex intensity decay. Under wind conditions with the same absolute wind speed, convective effects result in higher vortex intensity for negative wind speeds than positive wind speeds, leading to longer-lasting wake vortices. Moreover, the rate of vortex intensity decay is not linearly correlated with wind speed, as vortex intensity decays faster at wind speeds of 1-3 m/s compared to 4 m/s.

3.  With high vertical wind speeds, the vortex core is influenced by wind speed gradients and rotational rate gradients, which enhance the instability of the primary vortex core. This causes irregular movements and interactions within the vortex core, forming secondary vortex structures. Simultaneously, the horizontal position of the primary vortex core rapidly contracts inwards.

4.  Under low-wind-speed conditions, the vortex core briefly contracts inwards before stabilizing into oscillations. Vertical wind inhibits the reduction in vortex core spacing, resulting in a slower reduction rate compared to calm wind conditions. This leads to a larger wake turbulence hazard zone. Under high wind speed conditions, the vortex core initially contracts rapidly inwards before stabilizing slowly. The vertical movement of the vortex core is in the direction of the wind. The greater the wind speed, the greater the displacement. Notably, for wind speeds of 1 m/s and 2 m/s, the effect of vortex core descent is most significant. As wind speed increases to a certain threshold, the impact of core decline remains relatively constant due to the influence of airflow inertia effects.

**Supplementary Materials:** The following supporting information can be downloaded at: https://www.mdpi.com/article/10.3390/app14010086/s1.

**Author Contributions:** J.Y. and C.L. established the flow field model, divided the grid, and performed the grid independence test. J.L. debugged the numerical simulation conditions and performed numerical simulation. J.Y. and Z.Z. processed the data of the rear flow field and analyzed the results. C.L. provided the literature review and contributed to the writing of the manuscript. J.L. and Z.Z. reviewed the paper and supervised the work. All authors have read and agreed to the published version of the manuscript.

**Funding:** This research was supported by the National Key R&D Program of China (No. 2022YFB2602401), Safety Capability Fund of the Civil Aviation Administration of China (research on the optimization of the operation mode of the closely spaced runway of Hongqiao airport), Nanjing University of Aeronautics and Astronautics Graduate Research and Practice Innovation Program (xcxjh20230706).

**Institutional Review Board Statement:** Not applicable.

**Informed Consent Statement:** Not applicable.

**Data Availability Statement:** Data are contained within the Supplementary Material.

**Conflicts of Interest:** The authors declare no conflict of interest.

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
