# Peer review of "Numerical Study of Aircraft Wake Vortex Evolution under the Influence of Vertical Winds"

_applsci, doi:10.3390/app14010086_

Round 1

Reviewer 1 Report

Comments and Suggestions for Authors

This numerical study focuses on the evaluation of vortex structures downstream of a simplified wing model considering wind velocities in vertical direction.  Numerical simulations were performed on the simplified wing model instead of solving the complete aircraft model. The decay rate was investigated under various wind speeds. Decay rate was found to be trivial when the vertical speed is high, which is a very interesting outcome that can be important for the safe operation of aircrafts in aviation industry. Following major points need to be addressed first for further consideration of the outcomes:

1.       Abstract needs to be revised highlighting the simplified wing model (at line 13).    

2.       References should be given in a universal viewpoint rather than mentioning as domestic and foreign researchers.

3.       Mentioning experimental subject through the paper must be changed to numerical experiments or numerical simulations since no experimental study was presented in the study.

4.       Why the k-omega SST turbulence closure model was selected? Is there a certain reason to the presented problem?

5.       Why the temperature field was solved for the incompressible flow (Eqs. 1 and 2)? Is it required for the calculation of temperature dependent viscosity? If so, which viscosity model was used?

6.       Numerical schemes used for the discretizations of unsteady terms in the governing equations should be given along with the stability criteria.   

7.       How the equations were solved during the numerical simulations?

8.       At which platform were the simulations performed? Was parallel computing used during simulations?

9.       There is no validation for the numerical model? The numerical model must be validated with the experimental or numerical data from the literature. Otherwise, outcomes of the study will be questioned.

Reviewer 2 Report

Comments and Suggestions for Authors

1. In the introduction --> it is necessary to clarify the purpose of the research. No need to write a chapter. Because the paper is not a research report.
2. Line 164 and 165 --> Figure 2 and Table 1 are written reference sources.
3. In the meshing section (line 209) it should be explained the mesh variation and model variation as well as how the amount of meshing is approached using what.
3. Line 270 --> Boundary condition is too weak, it is necessary to add variation of inlet velocity and variation of the viscous model used.
4. In the discussion results section --> it is necessary to add comparative data from several previous studies (can be in the form of a table or graph).
5. The conclusion (line 521) should answer the purpose of the introduction
6. Line 569 --> added 2-3 papers in references.

Comments on the Quality of English Language

1. In the introduction --> it is necessary to clarify the purpose of the research. No need to write a chapter. Because the paper is not a research report.
2. Line 164 and 165 --> Figure 2 and Table 1 are written reference sources.
3. In the meshing section (line 209) it should be explained the mesh variation and model variation as well as how the amount of meshing is approached using what.
3. Line 270 --> Boundary condition is too weak, it is necessary to add variation of inlet velocity and variation of the viscous model used.
4. In the discussion results section --> it is necessary to add comparative data from several previous studies (can be in the form of a table or graph).
5. The conclusion (line 521) should answer the purpose of the introduction
6. Line 569 --> added 2-3 papers in references.

Round 2

Reviewer 1 Report

Comments and Suggestions for Authors

Thanks to the authors for revising the manuscript according to suggestions. The authors indicated that the numerical model is a coupled model. However, the code used in the study is still ambiguous. Please give details of the computer code such as commercial software, open-source code or an in-house code developed by the authors. 

Reviewer 2 Report

Comments and Suggestions for Authors

This paper can be published

Comments on the Quality of English Language

This paper can be published

Author Response

Thank you very much for your review and support. We are delighted that you have agreed to publish our manuscript. Once again, thank you for your time and assistance.